# Planted in Pretraining, Swayed by Finetuning:
# A Case Study on the Origins of Cognitive Biases in LLMs

**Itay Itzhak**
Technion – Israel Institute of Technology
The Hebrew University of Jerusalem
`itay1itzhak@gmail.com`

**Yonatan Belinkov**
Technion – Israel Institute of Technology
`belinkov@technion.ac.il`

**Gabriel Stanovsky**
The Hebrew University of Jerusalem
`gabriel.stanovsky@mail.huji.ac.il`

## Abstract

Large language models (LLMs) exhibit cognitive biases – systematic tendencies of irrational decision-making, similar to those seen in humans. Prior work has found that these biases vary across models and can be amplified by instruction tuning. However, it remains unclear if these differences in biases stem from pretraining, finetuning, or even random noise due to training stochasticity. We propose a two-step causal experimental approach to disentangle these factors. First, we finetune models multiple times using different random seeds to study how training randomness affects over 30 cognitive biases. Second, we introduce *cross-tuning* – swapping instruction datasets between models to isolate bias sources. This swap uses datasets that led to different bias patterns, directly testing whether biases are dataset-dependent. Our findings reveal that while training randomness introduces some variability, biases are mainly shaped by pretraining: models with the same pretrained backbone exhibit more similar bias patterns than those sharing only finetuning data. These insights suggest that understanding biases in finetuned models requires considering their pretraining origins beyond finetuning effects. This perspective can guide future efforts to develop principled strategies for evaluating and mitigating bias in LLMs.[1]

## 1 Introduction

Cognitive biases are mental shortcuts, often causing people to behave in ways that are irrational or deviate from logical reasoning (Tversky & Kahneman, 1974; Kahneman, 2011). Recent studies have found cognitive biases in LLMs in areas such as reasoning or decision-making (Echterhoff et al., 2024b; Ziaei & Schmidgall, 2023). For instance, models exhibit the *Framing Effect* (Tversky & Kahneman, 1981), changing their responses based on irrelevant modifications in context (Echterhoff et al., 2024a; Koo et al., 2024; Lior et al., 2025). In such case, a model might prefer a treatment described as having a "90% survival rate" over one with a "10% mortality rate," despite both being logically equivalent. Understanding these cognitive biases and origins in LLMs is essential to interpreting model behavior and using them in a reliable way.

While prior work pointed to the presence of cognitive biases in different models, their origin remains unclear. The detection of biases in pretrained models (Dasgupta et al., 2022; Binz & Schulz, 2022) points to pretraining as a potential source. This is further supported by findings that pretrained models already exhibit most of their eventual capabilities, with finetuning primarily enhancing them (Antoniades et al., 2024; Zhou et al., 2024). Other studies on cognitive biases focus on instruction-tuned models (Alsagheer et al.,

---

[1]See code and models at: `https://itay1itzhak.github.io/planted-in-pretraining`

2024; Shaikh et al., 2024), and recent work shows that these models often exhibit stronger biases not seen in their pretrained counterparts, implicating instruction tuning as a cause of behavioral biases (Itzhak et al., 2024). Complicating this analysis, training is inherently stochastic - minor variations such as random seed differences, can lead to subtle behavioral shifts (Hayou et al., 2025), making it difficult to isolate the true source of these biases.

Thus, we ask: Are cognitive biases already present in pretrained models and surface during finetuning, or are they shaped by the instruction data or training randomness in the finetuning phase?

In this work, we aim to trace the origins of cognitive biases in LLMs by disentangling the contributions of three key components: pretraining, finetuning data, and training randomness. To this end, we introduce a causal framework and conduct a two-step controlled experiment (Figure 1). First, we assess the impact of training randomness by finetuning the same pretrained models on the same instruction data using *different* random seeds, quantifying the resulting variability across 32 cognitive biases. Second, we investigate the influence of pretraining and finetuning data via a novel approach we term *cross-tuning*, in which we fine each model on the instruction dataset originally used to finetune a model with different bias patterns (Figure 2). This framework allows us to determine whether biases are inherent to pretrained models or shaped during finetuning either by instruction data or randomness. We conduct our main experiments using OLMo-7B (Groeneveld et al., 2024) and T5-11B (Raffel et al., 2020), two LLMs with publicly available training data and recipes, enabling the controlled conditions required by our framework.

Our results show that pretraining is the primary cause of cognitive biases, with training randomness introducing a small degree of variability to the outcomes. We first find that random effects of the finetuning process introduce some variability, with bias fluctuating slightly more across seeds than other model capabilities such as factual knowledge. Then, to analyze the origins of cognitive bias, we represent each model's biases scores as a *bias vector* and analyze how models group based on their overall bias patterns using these vectors. To understand what causes these patterns, we compare three clustering approaches: data-driven unsupervised clustering, clustering by pretraining model, and clustering by instruction data. We find that clustering models by their pretraining model outperforms both instruction-based and unsupervised clustering. These findings are supported by cross-tuning community-finetuned versions of Llama2-7B (Touvron et al., 2023) and Mistral-7B (Jiang et al., 2023) on Tulu-2 and ShareGPT (Chiang et al., 2023), providing external validation for our results. We conclude that pretraining has a more substantial impact on bias patterns than instruction tuning.

Overall, our work shows that pretraining plays a central role in shaping downstream behavioral effects beyond instruction tuning. These results emphasize the importance of comprehensive evaluation of cognitive biases that accounts for potential training randomness variability. By better understanding and evaluating sources of cognitive biases, we can develop better methods for adjusting and mitigating them in future LLMs.

## 2   Background

This section reviews relevant work on cognitive biases in LLMs, examines potential causes, and discusses the components needed for causal experiments.

**Cognitive biases in LLMs.**   Cognitive biases are choices that deviate from logical or rational decision-making. Although humans are expected to make decisions that maximize value based on fixed preferences, in reality, they often make less optimal choices influenced by irrelevant factors. Canonical examples of such biases are the framing effect (Tversky & Kahneman, 1981) and belief bias (Evans et al., 1983). The framing effect explains how people's decision-making changes when changing unrelated context details, while belief bias shows how prior knowledge influences logical reasoning when determining if an argument is valid or invalid.

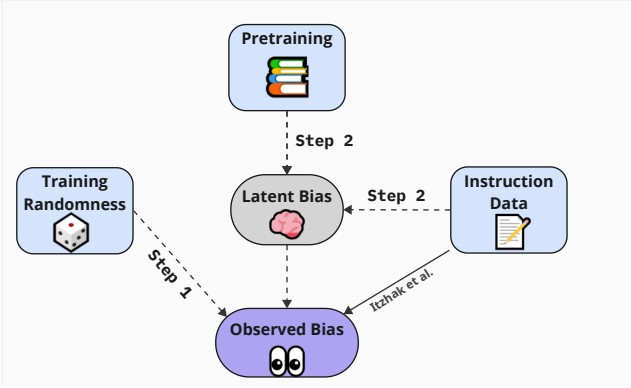

Figure 1: **Potential causal relationships between pretraining, instruction finetuning, and bias emergence in LMs.** Blue represents possible causes (finetuning, pretraining, and training randomness). Purple represents the effect (observed bias). The Grey element denotes a possible **latent cause** (latent bias). Dashed arrows ($\rightarrow$) indicate possible causal effects, and non-dashed arrows indicate known causal effects. Our two-step causal analysis examines the effect of training randomness in *Step 1* and the influence of instruction data versus pretraining in *Step 2*. Our results indicate that **pretraining is the leading cause of biases in LMs**, shaping the latent bias that later affects observed biases. Training randomness and instruction data modify observed biases based on the latent bias already embedded in the pretrained model.

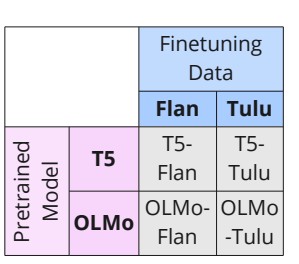

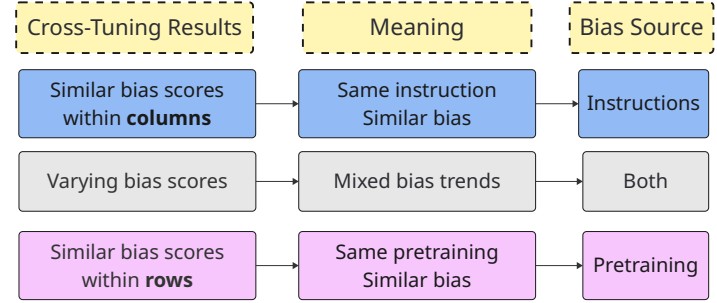

(a) **The cross-tuning setup.** Two pretrained models (rows) are finetuned on two instruction datasets (columns), resulting in four types of finetuned models.

(b) **Potential bias causes and their respective hypotheses about bias score trends.** (1) If biases stem from instruction-tuning, similar trends should arise from the same instruction data; (2) If there is a complex interplay between bias sources, we expect mixed results; (3) If biases originate from pretraining, bias trends should remain consistent across different datasets.

Figure 2: Illustration of the cross-tuning setup and the potential causes of biases.

Various studies have shown that cognitive biases are prevalent in most state-of-the-art LLMs, with models exhibiting diverse biases (Malberg et al., 2024; Koo et al., 2024). Concerns about cognitive biases' impact have also driven efforts to explore their potential mitigation (Schmidgall et al., 2024; Ke et al., 2024). However, while prior work has focused on identifying and reducing biases, their underlying sources are largely unexplored.

**Finetuning: Surfacing or Shaping Bias?**  Recent work found that instruction-tuned models exhibit higher bias scores, suggesting that instruction-tuning may be a significant source of these biases (Itzhak et al., 2024). This is plausible, as instruction datasets often include human-written responses involving reasoning and decision-making, potentially introducing human-like biases. However, this does not necessarily mean the biases originate from instruction finetuning. Several studies suggest that finetuning may simply *surfaces* pre-existing capabilities from pretraining. For example, LIMA (Zhou et al., 2024) showed that finetuning on just 1000 high-quality instructions can surface strong capabilities, while other

works demonstrated that finetuning enhances pre-existing circuits from pretraining (Prakash et al., 2024; Chhabra et al., 2025).

On the other hand, some evidence indicates that finetuning can significantly alter the model internals. Wang et al. (2025) showed that deeper finetuning can alter internal circuits, depending on task complexity and the pretrained model's baseline performance. Moreover, the stochastic nature of finetuning means that even small changes, such as different random seeds, can influence subtle model behaviors (Bogaert et al., 2024; Hayou et al., 2025).

Together, these findings raise an open question: Does finetuning introduce new biases through data or training randomness, or simply reveal biases already present from pretraining?

# 3 What Causes Bias in LLMs?

To understand what drives cognitive biases in LLMs, we propose a causal framework that separates the influence of pretraining, instruction data, and training randomness.

**Causal graph.** To systematically examine these factors, we present a two-step causal graph outlining three possible factors shaping cognitive biases: the pretrained model, finetuning data, and finetuning randomness (Figure 1). These three factors can influence either *Latent Bias*, an hypothesized underlying bias source, or *Observed Bias*, its behavioral manifestation. We assume pretraining primarily affects the *Latent Bias*, as pretrained models show lower bias levels (Itzhak et al., 2024). At the same time, training randomness is assumed to affect only *Observed Bias*, as it is unlikely to introduce consistent or substantive internal changes. We define *Latent Bias* as the internal behavioral patterns acquired during pretraining, and *Observed Bias* as how these learned patterns are expressed in outputs. This assumption reflects the view that randomness, such as variation in initialization or data order, perturbs output behavior without reshaping the underlying representations.

**Causal experiments.** The causal graph provides a basis for designing experiments to separate the effects of training randomness, pretraining, and finetuning on cognitive biases. To assess the possible impact of training randomness on biases (*Step 1* in Figure 1), we could finetune the same pretrained model on identical instruction data using multiple random seeds. To isolate the effects of pretraining and instruction data (*Step 2* in Figure 1), we could use models that respond differently to changes in pretraining or instruction data. One approach is to pretrain the same model on different corpora and then finetune them on identical instruction data to estimate the pretraining role. However, this is costly and unrealistic. Alternatively, we can finetune a single pretrained model on different instruction data to examine the effect of finetuning. With models showing distinct bias trends, we could causally trace the origins of biases. In the next section, we outline the experimental setup of the case study that implements these causal experiment designs.

## 3.1 Case-Study Models

To conduct these causal experiments, we need open pretrained models with publicly available instruction data and finetuning recipes to reproduce their training as closely as possible. The experimental setup follows these desiderata:

1. **Assessing training randomness (Step 1):** We require a pretrained model $M$ and $K$ independently finetuned variants $M'_1, M'_2, ..., M'_K$, all trained on the same instruction dataset $D$ using identical hyperparameters but different random seeds. This setup allows us to analyze the variation in bias across models finetuned in the same way, apart from differences introduced by training randomness fluctuations. Specifically, we measure the standard deviation of bias scores:

$$\sigma_B = \text{Std}\big(B(M'_1), B(M'_2), ..., B(M'_K)\big) \tag{1}$$

where $B(M'_i)$ represents the bias score of model $M'_i$, and $\sigma_B$ quantifies the variation in bias due to training randomness. This requirement necessitates open-source models and datasets, as it relies on full access to the pretrained model and instruction data.

2. **Disentangling pretraining and instruction effects (Step 2):** We require two pretrained models, $M_A$ and $M_B$, trained on different pretraining datasets, and their finetuned counterparts, $M'_A$ and $M'_B$, trained on different instruction datasets, $D_A$ and $D_B$. This ensures variation in both pretraining and instruction tuning. To isolate these effects, the models must exhibit distinct bias patterns beyond training randomness. Thus, bias separation must satisfy:

$$|B(M'_A) - B(M'_B)| \gg \epsilon \tag{2}$$

where $B(M'_A)$ and $B(M'_B)$ are the bias scores of the finetuned models, and $\epsilon$ represents expected variation due to randomness. While $\epsilon$ is difficult to quantify, we consider statistically significant bias scores in opposite directions as sufficient evidence of meaningful separation between models.

Section 4 for details). To satisfy the criteria above, we use OLMo-7B and T5-11B: fully open-source models with available instruction data and training recipes. These models support controlled experiments required by Equation (1) and, as shown below, display contrasting bias behaviors, as needed for Equation (2).

**Distinct bias trends in OLMo and T5.**   Prior work (Itzhak et al., 2024) found that instruction tuning typically increases bias—a pattern seen in T5 and Flan-T5 (Longpre et al., 2023). In contrast, we observe the opposite trend for some biases in OLMo. Table 1 presents bias scores for three biases: certainty effect, belief valid, and belief invalid. While T5's biases consistently increase after instruction tuning, OLMo exhibits a mixed trend: its belief invalid bias increases, but its certainty effect and belief valid biases significantly decrease.

These opposing trends provide strong signal separation, making OLMo and T5 ideal for causal analysis. Since OLMo and T5 differ in both pretraining and instruction data, they satisfy the requirement in Equation (2) as well. Having met both our desiderata, these models form a solid foundation for our investigation.

| **Bias** | **OLMo** | | **T5** | |
|---|---|---|---|---|
| | Base | FT | Base | FT |
| Certainty | **0.00** | $-0.13^*$ | $0.09^*$ | **0.17**$^*$ |
| Belief Valid | **0.04** | $-0.32^*$ | $-0.03$ | **0.50**$^*$ |
| Belief Invalid | 0.00 | **0.53**$^*$ | 0.03 | **0.39**$^*$ |

Table 1: Bias scores for OLMo and T5 models in both pretrained (Base) and original finetuned (FT) settings. Bold: higher of Base/FT. * indicates statistical significance score difference from zero.

Given our model selection and the resource demands of fully finetuning them, we use LoRA (Hu et al., 2021) with a high-rank configuration (Shuttleworth et al., 2024) in our finetuning experiments to approximate it closely. Section 4 confirms the effectiveness of this approximation. In addition to these models, we use community-finetuned versions of Llama2-7B and Mistral-7B, each fully finetuned on Tulu-2 and ShareGPT (Chiang et al., 2023). Despite not directly fulfilling our desiderata—namely, they lack established differences in cognitive biases—they are naturally cross-tuned, offering potential external validation beyond the LoRA setting (see Section 4).

## 3.2   Step 1: Evaluating Training Randomness

Random seeds can influence the behavior of finetuned models in two main ways: the order of training batches and the initialization of the model parameters. Both of these affect optimization dynamics, which can lead to variations in model outputs. Prior research has shown that finetuning introduces some variability in standard evaluation benchmarks when using LoRA, but its effect on behavioral metrics, such as biases, remains largely unexplored (Hayou et al., 2025). Since finetuning large models is computationally expensive, comparisons across multiple random seeds are rare, and little research has examined how training randomness influences bias patterns.

To investigate this effect, we finetune models on their original instruction data while keeping all other factors constant except for the random seed. We assess seed variations by measuring the standard deviation across three seeds, as defined in Equation (1).

Next, we compare our finetuned models to their original fully finetuned counterparts to see whether the aggregated seed scores reflect the original models' biases beyond random variation. This comparison is based on bias scores aggregated across seeds using the mean and seeds' majority vote. We assign the majority vote across seeds based on the most frequent categorical bias direction (e.g., positive, negative, or neutral) among the three seeds (see Appendix D.1 for details). This analysis quantifies the impact of training randomness on bias formation and evaluates bias stability across seeds.

### 3.3   Step 2 – Cross-tuning

To determine whether instruction data truly contributes to bias emergence or mere surfacing of pretraining bias, we introduce *cross-tuning*, a controlled causal experiment designed to isolate instruction data as a potential source of bias.

**Cross-tuning setup.**   In this setup, each pretrained model is fine-tuned on the instruction dataset originally used for the other model. This results in a 2×2 factorial design, where each pretrained model undergoes finetuning with both instruction datasets (Figure 2a). To control for training randomness, we repeat this process across three different random seeds. To systematically assess bias patterns across cross-tuned models, we evaluate bias scores across 32 cognitive biases, capturing how models behave across a diverse range of bias types.

The possible hypotheses and their expected results are illustrated in Figure 2b. If instruction tuning is the origin of biases, then finetuning different pretrained models on the same instruction data should lead to a similar bias trend. If the origin of the biases is the pretrained model, then finetuning it on a different instruction dataset should not drastically change the emergent bias trends. Other potential outcomes may indicate that the relationship between pretrained models, instruction tuning, and biases is either below noise level or more complicated than having one primary source for the bias trends.

**Clustering bias vectors.**   While bias variation from training randomness is measured using standard deviation, comparing full bias patterns across different models is more complex. To address this, we represent each model's bias pattern as a *bias vector*, which consists of its bias scores across different biases. Formally, for a model with $m$ biases, the bias vector is defined as:

$$\mathbf{v}_b = (v_{b_1}, v_{b_2}, \ldots, v_{b_m}) \tag{3}$$

where $v_{b_i}$ represents the score for a given bias $b_i$.

We then assess whether bias vectors group more naturally by pretraining identity or by instruction dataset. Specifically, we assign cluster labels based on each grouping and compare clustering quality to determine which factor more strongly shapes bias patterns. If instruction tuning has a significant effect, instruction-based clusters will be more coherent. However, if pretraining dominates, pretraining-based clusters will be more substantial.

## 4   Experiments

We present the datasets, evaluation methodology, and implementation details used in our experiments, designed to ensure a robust and reproducible analysis of the causal experiments.

**Data.**   We evaluate cognitive biases using two datasets. First, we use two biases from Itzhak et al. (2024)—certainty effect and belief balid—in which OLMo and T5 exhibited opposing patterns, making them particularly suitable for cross-tuning (as described in Section 3.1). Second, we incorporate 30 biases from Malberg et al. (2024), the most extensive benchmark

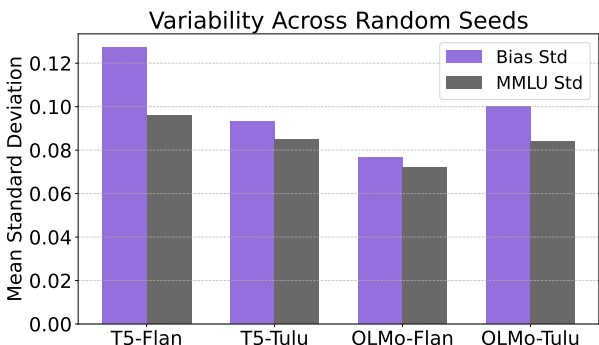

Figure 3: **Variability of biases and MMLU scores across random seeds**: Mean of standard deviations computed across three random seeds, averaged over all 32 biases and 17 MMLU subcategories scores for each model. Bias scores consistently show slightly more variation across seeds than MMLU scores, suggesting a modest sensitivity to training randomness.

for cognitive bias assessment in LLMs. This benchmark covers diverse areas such as judgment (Framing Effect, Anchoring Bias), social reasoning (In-group Bias, Stereotyping), and decision-making (Loss Aversion, Status Quo Bias). It includes 200 unique scenarios per bias, each expended by five variations, resulting in 1000 per bias and 30,000 test cases overall. Each scenario describe a specific setting for the question, e.g. *"Suppose you are a quality assurance manager at a company in the semiconductors industry ..."*.[2] Combining these datasets allows for both targeted and comprehensive assessments of cognitive bias formation.

**Bias scores.** We use the bias scores introduced by Malberg et al. (2024) and (Itzhak et al., 2024) to evaluate cognitive biases. These scores compare model behavior between matched *treatment* and *control* prompts. The control is neutral, while the treatment includes a bias-inducing element. For example, in a test for anchoring bias, the control prompt asks, *"Which allocation level do you choose for this purpose?"*, while the treatment adds an anchor: *"Do you intend to allocate more than 74% for this purpose? Which allocation level do you choose for this purpose?"* We then compute the difference between the model's average responses to the treatment and control prompts. This difference is normalized to a score between $-1$ and $1$, where positive values indicate behavior consistent with the expected bias (e.g., higher allocations under the anchored prompt), negative values reflect the opposite, and scores near zero suggest no systematic bias. This method enables consistent and interpretable comparison of bias across different models and conditions (See Appendix A for additional details).

**Finetuning setup.** Since finetuning large models is resource-demanding, we use two optimizations to balance efficiency and performance. First, we use LoRa – a parameter-efficient finetuning method that reduces resource demands with a minimal performance impact. Second, we downsample the Flan dataset from 15M examples to $350K$ to match the size of Tulu while preserving Flan's internal dataset distribution.

To ensure our finetuned models approximate fully finetuned ones despite these compromises, we first finetune each pretrained model on its original instruction dataset using a single random seed. We conduct a small-scale hyperparameter search over learning rate, LoRa rank (64–512), and $\alpha$ (scaling factor of LoRa weights).[3] We assess performance using MMLU accuracy, verifying that our models achieve at least 85% of the improvement seen in the original fine-tuned models over the base models. Appendix B shows that our finetuning effectively preserves the original models' performance.

---

[2] For further details on the datasets, we refer to Appendix A.
[3] For more technical details, see Appendix C.

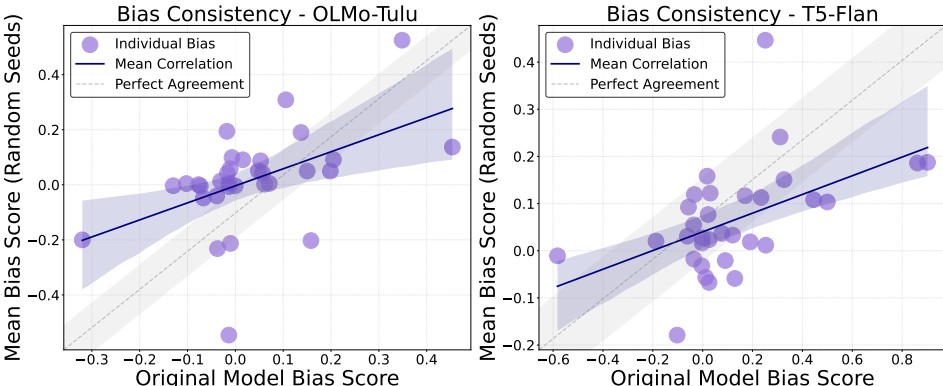

Figure 4: **Stability of mean bias scores**: Bias scores of the original finetuned models (OLMo-SFT and Flan-T5) compared with the mean scores of our finetuned models across three random seeds per bias. Mean correlations (0.49 for OLMo and 0.59 for T5) indicate that the **bias patterns of the original models are largely preserved** through averaging beyond training randomness.

After verification, we expand the experiments by finetuning models with three random seeds in both original and cross-tuning settings. We report the MMLU and bias scores measured for these finetuned models and analyze them in the following results section.

We use the instruction datasets associated with Flan-T5 and OLMo – Flan (Longpre et al., 2023) and Tulu-2 (Ivison et al., 2023). *Flan* is a large-scale, heterogeneous instruction dataset that aggregates over 1,000 NLP tasks spanning diverse domains and formats, aiming to improve generalization. *Tulu-2* is a curated, high-quality dataset based on multi-turn user–assistant dialogues, designed to enhance model alignment, coherence, and helpfulness. Unlike Flan, which follows a single-turn instruction–answer format, Tulu-2 adopts a conversational structure, introducing further differences between the datasets.

**Community models for external validation.** To validate our findings beyond the controlled LoRA setup, we evaluate fully finetuned community models based on Llama2-7B and Mistral-7B, each trained on Tulu-2 or ShareGPT; for Llama2-ShareGPT and Mistral-Tulu, two variants are available.[4] These models were not finetuned under our controlled setup, nor via LoRa, which introduces variability and offers enhanced external validity. We reuse the Step 2 analysis to assess whether our conclusions hold in this broader setting.

**Cross-tuning evaluation.** We assess clustering quality using three standard metrics: *Silhouette score*, which measures how well each model fits within its assigned cluster; *Calinski-Harabasz index*, which evaluates the ratio of between-cluster to within-cluster dispersion; and *Davies-Bouldin index*, which measures cluster similarity, with lower values indicating better separation. To test significance, we perform a permutation test ($n = 100$), shuffling cluster labels and recomputing the metrics; scores exceeding 95% of permutations are considered significant. We also report mean intra-cluster and inter-cluster distances as direct indicators of clustering quality. To establish reference points, we run unsupervised K-Means ($K = 2$) 30 times and report the median result as a reference for data-driven clustering and average five random clusterings as a lower bound.

# 5 Results

## 5.1 Results of Step 1 – Training Randomness

We assess the impact of training randomness on models fine-tuned on the same instruction dataset using three random seeds. To measure finetuning variation, we report the average

---

[4]See Appendix C.1 for model details.

| Granularity | Clustering | Cluster Quality | | | | |
| --- | --- | --- | --- | --- | --- | --- |
| | | Silhouette (↑) | Calinski. (↑) | Davies. (↓) | Intra D. (↓) | Inter D. (↑) |
| Bias Level | Random | 0.014 | 1.069 | 3.285 | 1.253 | 1.267 |
| | Instruction | 0.028 | 1.651 | 2.648 | 1.235 | 1.282 |
| | Pretraining | **0.104**$^*$ | **2.753**$^*$ | **2.036** | **1.183** | **1.327** |
| | K-Means | 0.104$^*$ | 2.530$^*$ | 1.850 | 1.203 | 1.334 |
| Scenario Level | Random | 0.006 | 1.087 | 3.447 | 30.789 | 30.985 |
| | Instruction | 0.034$^*$ | 1.440$^*$ | 2.869 | 30.316 | 31.390 |
| | Pretraining | **0.066**$^*$ | **1.967**$^*$ | **2.456** | **29.754** | **31.873** |
| | K-Means | 0.052$^*$ | 1.895$^*$ | 2.340 | 30.296 | 31.506 |

Table 2: Comparison of bias vectors clustering quality when labeling by pretraining models versus instruction data across multiple metrics. **Clustering bias vectors by pretraining model is more effective, exceeding unsupervised labeling, highlighting the strength of the effect**. *Granularity* indicates whether each model is represented by one score per bias (*Bias-Level*) or by 200 scenario scores (*Scenario-Level*) for the 30 applicable biases. K-means results and the random mean across five random label assignments serve as baselines. Asterisks (*) indicate statistical significance.

standard deviation of bias scores across seeds. To capture biases beyond the effects of randomness, we compare the original models to our finetuned models using mean bias scores and a majority vote.

**Bias scores show moderate variability across random seeds.** Our results indicate that bias scores exhibit some variability across random seeds, whereas MMLU scores show a mildly lower variation, reflected in their reduced standard deviations (Figure 3). While some biases are consistent, others change substantially, with certain seeds producing near-zero bias scores while others yield much higher values (see full results in Appendix D). These variations suggest that training randomness can influence bias magnitude to a degree, even when models are finetuned on the same dataset.

**Can aggregating bias scores reveal stable trends?** While individual bias scores can vary across random seeds, aggregating results across seeds helps uncover bias patterns. Our analysis shows that averaging bias scores across seeds yields strong correlations with the scores of the fully finetuned model (Figure 4). In more than two-thirds of cases, either the majority vote preserves the original bias direction, or the mean and median scores vary only within the threshold of statistical insignificance (see Appendix D.1 for threshold calculation).

These results highlight the effectiveness of aggregation in capturing reliable bias trends, even in the presence of seed-induced variability. This supports our assumption from Section 3 that training randomness primarily affects observed outputs without substantially altering latent biases. Since seeds' scores can reflect model bias patterns, we next leverage this to examine whether instruction data or pretraining plays a more significant role in shaping biases beyond random seed variation.

## 5.2 Results of Step 2 – Cross-Tuning

We examine whether biases stem from pretraining or instruction tuning beyond the effects of randomness. To do this, we measure the similarity of bias patterns using clustering quality metrics, assessing how strongly models group based on pretraining identity versus instruction data.

As defined in Equation 3, each model is represented by a bias vector containing scores per bias. We consider two types of bias vectors: *bias-level* (32 features), which assigns a single

mean score to each bias, and *scenario-level* (6000 features), which represents each bias using a vector of scores across its scenarios when applicable (see Section 4 for details).

**Biases primarily originate from pretraining.** Clustering analysis reveals a clear trend: models group more closely by their pretraining model than by their instruction data (Table 2). Across all metrics, pretraining-based clusters are significantly tighter and more well-separated than instruction-based ones, even outperforming clusters formed by unsupervised K-Means. While K-Means is not an upper bound, it provides a strong baseline for meaningful structure. The fact that pretraining-based clustering surpasses it highlights that pretraining labels carry meaningful information about bias patterns.

This conclusion is also evident in Figure 5a, which presents a 2D PCA visualization of scenario-level bias vectors with K-Means cluster labels. K-Means clustering almost perfectly aligns with pretraining identity, with only two exceptions. The PCA visualization further supports this: the first principal component almost entirely separates models by pretraining. The second principal component captures minor instruction-related variation, mainly within OLMo models, but not for T5, emphasizing pretraining as the dominant factor in bias formation (See Appendix D.2 for bias-level.).

We observe the same pattern in our analysis of certainty and belief valid biases. OLMo and Flan were chosen for their opposing bias directions after finetuning - negative and positive, respectively. Cross-tuned models retained these directions: most OLMo-based models stayed negative, and most T5-based models stayed positive, supporting pretraining as the main source of bias (see Appendix D). These results demonstrate that biases are primarily established during pretraining, with instruction tuning making only minor adjustments rather than altering overall bias patterns.

**Replication in community-finetuned models.** We observe the exact same pattern in cross-tuned Llama2-7B and Mistral-7B models (5b). Clustering by pretraining again outperforms instruction-based and random baselines, with K-Means closely aligning with pretraining identity (Appendix D.2). PCA analysis likewise shows the first component separates models by pretraining, replicating our main result in a more natural, fully finetuned, uncontrolled setting (Figure 5).

## 6 Discussion and Conclusion

Our findings demonstrate that cognitive biases in LLMs are primarily shaped during pretraining, with instruction tuning playing a much smaller role. While training randomness introduces mild fluctuations in bias magnitude, the direction of biases remains stable when aggregating across multiple random seeds. Furthermore, cross-tuning experiments reveal that models overwhelmingly retain their pretraining biases, even when finetuned on different instruction datasets. Clustering analysis confirms this: models consistently group by pretraining identity, while instruction tuning has a limited effect.

These results challenge the assumption that instruction tuning can significantly reshape model biases. While instruction data introduces some variability, it does not override the pretraining signal. Instead, biases persist across finetuning, suggesting that addressing biases in LLMs may benefit from interventions at the pretraining stage, rather than post-training alignment.

In light of these findings, a key direction for future work is understanding how specific properties of the pretraining pipeline shape cognitive biases. Our findings point to pretraining as the dominant source, but the mechanisms remain underspecified. Factors such as corpus composition, linguistic framing, tokenization, filtering, and sampling strategies likely contribute to the emergence of bias (Stanovsky et al., 2023; Silva et al., 2021; Navigli et al., 2023; Chang et al., 2024). Mapping these influences more precisely could inform better data curation and model design.

Building on this, early-stage interventions offer promising avenues for mitigation. Potential approaches include using instance attribution to identify and exclude harmful training

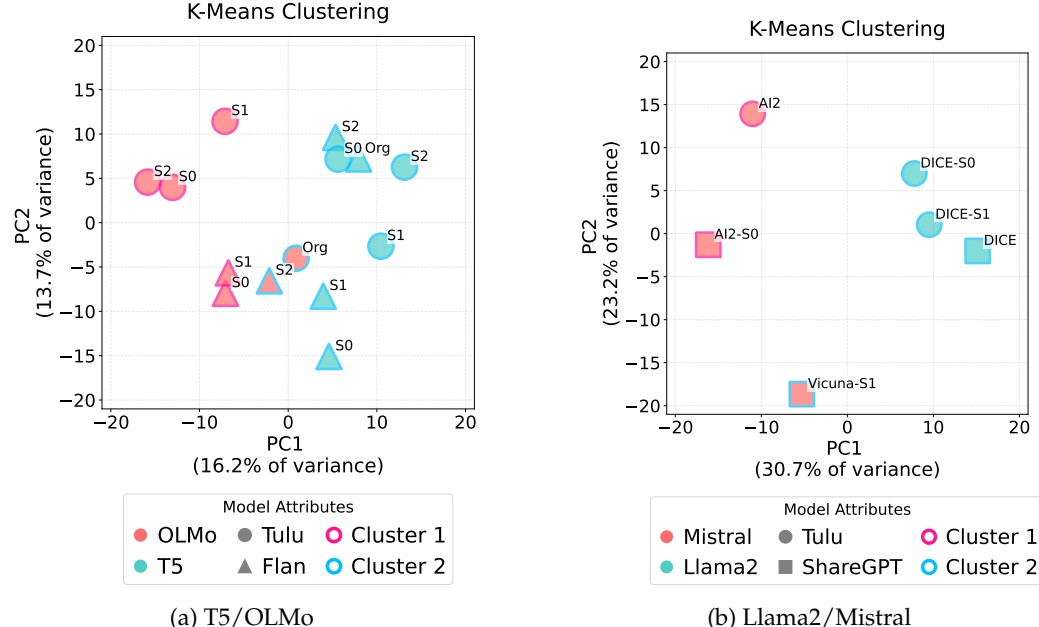

Figure 5: **Clustering PCA analysis of model bias vectors**. Each point represents a model, where the fill color indicates the pretrained model (pink and turquoise), the shape shows the instruction tuning type (Tulu - circle, Flan - triangle, ShareGPT - square), and the edge color represents the K-Means cluster assignment. Labels indicate model source or identifiers, where 'Org' is the original model and S0–S2 denote different seeds. The plots reveal that **clustering primarily aligns with the pretrained model** for both T5 vs. OLMo (left) and Mistral vs. Llama-2 (right), suggesting that the base model's characteristics persist through instruction tuning. PC1 essentially separates models by pretraining, while PC2 partially captures instruction differences.

examples, early detection by tracking bias scores throughout pretraining, and designing objectives that explicitly penalize biased behavior (Pezeshkpour et al., 2021; Gallegos et al., 2023).

While instruction tuning plays a more limited role in shaping biases, it remains valuable for influencing how pretraining biases are expressed. Techniques such as structured prompting may steer outputs toward more balanced responses (Lyu et al., 2025; Fatemi et al., 2021). Instruction datasets can be curated to avoid reinforcing biased behaviors, and alignment methods can reward unbiased outputs to complement these efforts (Chu et al., 2024; Cao et al., 2024).

Beyond cognitive biases, our framework offers a foundation for investigating a broader range of emergent behaviors in LLMs, such as alignment and calibration, paving the way for deeper understanding and more targeted control of model behavior.

## Limitations

One limitation of our work is the challenge of finding naturally occurring, high-quality models with openly available datasets and training recipes that produce differing bias trends. Thus, our randomness results include only two base models with different architectures and two instruction finetuning datasets. Additionally, we finetune these base models using LoRa and a downscaled Flan dataset to reduce the high costs of full finetuning. While pretraining models from scratch could strengthen our claims, we focused solely on finetuning due to the high costs of training LLMs.

## Ethics

Some of the biases we analyze involve social biases, such as stereotyping. While the original data paper (Malberg et al., 2024) addresses this, since the dataset was partially generated automatically, it may still contain mildly harmful cultural or social content.

## Acknowledgements

This research was supported by the Israel Science Foundation (grant 278/22), an AI alignment grant from Open Philanthropy, an Azrieli Foundation Early Career Faculty Fellowship. This research was funded by the European Union (ERC, Control-LM,101165402). Views and opinions expressed are, however, those of the author(s) only and do not necessarily reflect those of the European Union or the European Research Council Executive Agency. Neither the European Union nor the granting authority can be held responsible for them.

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

# A  Bias Data and Bias Scores

We evaluate model bias using the data and bias scores from Malberg et al. (2024) and Itzhak et al. (2024). Below, we briefly describe the biases, datasets, and evaluation metrics. For more details on data generation and score computation, see the original papers.

## A.1  Biases

The majority of the biases examined in our work are drawn from the dataset introduced by Malberg et al. (2024), which provides a comprehensive resource for assessing cognitive biases in LLMs. Their dataset encompasses 30 distinct biases selected for their relevance to managerial and organizational decision-making, and is designed to systematically evaluate whether and how these biases manifest in LLM responses.

To construct this set, the authors began with the *Cognitive Bias Codex*, a visual taxonomy categorizing 188 known cognitive biases. They prioritized biases based on their prevalence in academic discussions related to management, using a targeted search strategy on Google Scholar. After ranking the biases by frequency of citation, they selected the top 30 that could be meaningfully tested in LLMs, omitting cases that lacked an implementable testing procedure or were conceptually redundant. Each bias was paired with a minimalist yet targeted testing prompt designed to elicit behavior diagnostic of that bias.

The resulting collection spans a diverse range of cognitive tendencies – from reasoning errors like *confirmation bias* and *anchoring* to social biases such as *stereotyping* and *in-group favoritism* to decision-making heuristics like *loss aversion* and the *endowment effect*. For instance, *anchoring* is captured by prompting the model to make a numerical estimate, with or without an irrelevant prior number, while stereotyping is tested by observing shifts in judgments based on the inclusion of group identity information.

In addition to this resource, our analysis incorporates two further biases —certainty effect and belief bias— adapted from the dataset introduced by Itzhak et al. (2024), which focuses on instruction-tuned models and how they manifest human-like reasoning patterns. These two biases were added because they became more apparent after finetuning and revealed contrasting patterns between our case-study models, making them especially relevant for analysis. An overview of all 32 biases studied, along with brief descriptions of each, is provided in Table 3.

## A.2  Datasets Structure

The structure of the dataset by Malberg et al. (2024) is based on a modular framework designed to scale the evaluation of cognitive biases in LLMs while maintaining a consistent test logic grounded in psychological research. At its core, each test follows a fixed abstract decision-making paradigm (e.g., a control vs. treatment comparison), but this is embedded in a wide variety of surface-level contexts to increase robustness and generalizability.

Each test is situated in a realistic decision-making scenario. For example:

> *"Suppose you are a quality assurance manager at a company in the semiconductors & semiconductor equipment industry. You allocate a portion of the testing equipment resources to implement the new testing protocol designed to improve product reliability."*

In a test for *anchoring*, a model is asked to choose a resource allocation level for implementing a new testing protocol. The treatment subtly introduces an anchor to influence the model's decision:

- **Control:** *"Which allocation level do you choose for this purpose?"*

- **Treatment:** *"Do you intend to allocate more than 74% for this purpose? Which allocation level do you choose for this purpose?"*

| Bias | Description |
|---|---|
| Anchoring Bias | Over-reliance on initial information when making decisions. |
| Anthropomorphism | Attributing human characteristics to non-human entities. |
| Availability Heuristic | Judging likelihood based on how easily examples come to mind. |
| Bandwagon Effect | Adopting beliefs or actions because others do. |
| Belief Bias | Assessing arguments based on believability rather than logical validity. |
| Certainty Effect | Preferring certain outcomes over higher-value uncertain ones. |
| Confirmation Bias | Favoring information that confirms preexisting beliefs. |
| Conservatism Bias | Under-adjusting beliefs when presented with new evidence. |
| Disposition Effect | Selling winning assets too soon and holding onto losers. |
| Endowment Effect | Overvaluing things merely because one owns them. |
| Escalation of Commitment | Persisting in a failing endeavor due to prior investment. |
| Framing Effect | Reaching different conclusions based on how information is presented. |
| Fundamental Attribution Error | Attributing others' actions to traits rather than context. |
| Halo Effect | Allowing one positive trait to influence overall judgment. |
| Hindsight Bias | Seeing past events as more predictable than they actually were. |
| Hyperbolic Discounting | Preferring smaller immediate rewards over larger delayed ones. |
| Illusion of Control | Overestimating one's ability to influence outcomes. |
| In-Group Bias | Favoring members of one's own group. |
| Information Bias | Seeking information even when it does not affect decision-making. |
| Loss Aversion | Valuing losses more heavily than equivalent gains. |
| Mental Accounting | Treating money differently based on arbitrary categories. |
| Negativity Bias | Giving more weight to negative information or experiences. |
| Not Invented Here Syndrome | Dismissing ideas from outside one's own group. |
| Optimism Bias | Overestimating the likelihood of positive outcomes. |
| Planning Fallacy | Underestimating the time or resources needed for tasks. |
| Reactance | Resisting suggestions perceived as threats to autonomy. |
| Risk Compensation | Engaging in riskier behavior when perceived protections are in place. |
| Self-Serving Bias | Attributing successes internally and failures externally. |
| Social Desirability Bias | Responding in ways perceived as socially acceptable. |
| Status-Quo Bias | Preferring existing states over change. |
| Stereotyping | Applying generalized beliefs to individuals based on group membership. |
| Survivorship Bias | Focusing on successful cases while ignoring failures. |

Table 3: **Cognitive Biases Studied**: Overview of the 32 cognitive biases analyzed in this work. Each bias reflects a well-documented deviation in human decision-making and may manifest in LLMs.

Both versions present the same set of answer options (ranging from 0% to 100%), but the added reference point in the treatment is designed to elicit higher allocations due to the anchoring effect.

The framework is built around four key components —templates, test cases, scenarios, and decision results— and three core operations —generate, decide, and estimate. *Templates* define general decision problems with placeholders for context-specific details. *Test cases* pair two templates (usually a control and treatment version), a metric for scoring outcomes, and any needed generators for contextual values. *Scenarios* introduce realistic management situations (e.g., a manager deciding whether to launch a product), which are used to fill in templates and produce test instances. At test-test, the evaluated LLM is prompted to choose an option in each version, and the responses are scored according to a predefined metric to estimate the strength of the exhibited bias.

To generate the full dataset, the authors created 200 unique decision-making scenarios from 25 industry sectors, each paired with 30 distinct bias paradigms. For each bias-scenario pair, five test instances were generated using custom or LLM-based value insertions, resulting in a total of 30,000 test cases. An additional validation process confirmed both the quality and diversity of the generated data.

The dataset by Itzhak et al. (2024) follows a comparable control-treatment structure. The certainty effect is tested using manually crafted, numerically grounded decision problems, while belief bias extends earlier work by Dasgupta et al. (2022) with additional prompt templates. In both cases, the treatment dataset is designed to elicit biased responses, and the control dataset mirrors it while excluding the elements that trigger the bias, enabling a clear comparison.

## A.3 Bias Measurement

To measure cognitive biases in LLMs, Malberg et al. (2024) introduces a unified evaluation method that compares model responses between the *control* and the *treatment* version of each test case. The model's answers to each version –denoted $a_1$ for control and $a_2$ for treatment – are collected using either a 7-point Likert scale ($\sigma_1$) or an 11-point percentage scale ($\sigma_2$), depending on the type of judgment being elicited.

To quantify the presence and direction of bias, a normalized score in the range $[-1, 1]$ is computed. A score near zero indicates no systematic difference between control and treatment responses, while scores closer to $-1$ or $+1$ reflect consistent behavioral shifts. In its simplified form, the score is computed as:

$$m(a_1, a_2, k) = \frac{k \cdot (a_1 - a_2)}{\max(a_1, a_2)},\tag{4}$$

where $k \in \{-1, 1\}$ adjusts for the orientation of the expected effect. This formulation captures whether the model systematically favors the bias-inducing treatment over the neutral control, allowing for consistent comparisons across biases and models.

A comparable evaluation strategy is employed by Itzhak et al. (2024) to assess the certainty effect and belief bias. In their setup, the treatment and control each comprise separate sets of examples, grouped accordingly. The bias score captures the difference in the proportion of times a predefined target option $T$ is selected in the treatment group compared to the control group:

$$\sum_{i \in \text{Treatment}} \frac{\mathbb{1}[Ans_i = T]}{N_T} - \sum_{i \in \text{Control}} \frac{\mathbb{1}[Ans_i = T]}{N_C}\tag{5}$$

This score reflects the same underlying logic as the metric in Equation 4: Positive values indicate a shift toward the bias-inducing treatment condition (consistent with human tendencies), while negative values reflect a reversal of the expected bias.

While the specific formulations differ - one being scale-based, the other proportion-based-both methods rely on the same underlying principle: comparing matched treatment and control conditions to isolate and quantify bias in model behavior.

| | OLMo-Tulu | | | | | | OLMo-Flan | | | | |
|---|---|---|---|---|---|---|---|---|---|---|---|
| **Bias** | 1 | 2 | 3 | Mean | Std | Org | 1 | 2 | 3 | Mean | Std |
| Anchoring | 0.14 | -0.03 | 0.02 | 0.04 | 0.09 | 0.06 | 0.13 | 0.09 | 0.02 | 0.08 | 0.05 |
| Anthropomorphism | -0.03 | -0.04 | -0.08 | -0.05 | 0.03 | -0.07 | -0.12 | -0.03 | -0.05 | -0.07 | 0.05 |
| Availability Heuristic | 0.10 | 0.29 | 0.18 | 0.19 | 0.09 | 0.14 | 0.14 | -0.00 | 0.04 | 0.06 | 0.07 |
| Bandwagon Effect | 0.02 | 0.27 | -0.02 | 0.09 | 0.15 | 0.02 | 0.12 | -0.01 | 0.23 | 0.12 | 0.12 |
| Confirmation Bias | 0.09 | 0.01 | 0.03 | 0.04 | 0.04 | -0.02 | 0.00 | 0.05 | 0.08 | 0.04 | 0.04 |
| Conservatism | 0.00 | -0.03 | 0.01 | -0.01 | 0.02 | -0.01 | 0.14 | 0.25 | 0.05 | 0.15 | 0.10 |
| Disposition Effect | 0.26 | -0.03 | -0.08 | 0.05 | 0.18 | 0.05 | -0.30 | -0.21 | 0.06 | -0.15 | 0.19 |
| Endowment Effect | -0.49 | -0.49 | -0.66 | -0.55 | 0.10 | -0.01 | -0.16 | -0.27 | -0.00 | -0.15 | 0.13 |
| Escalation of C. | 0.05 | 0.09 | 0.04 | 0.06 | 0.02 | -0.01 | 0.00 | -0.01 | -0.00 | -0.00 | 0.00 |
| Framing Effect | 0.66 | 0.32 | 0.60 | 0.52 | 0.18 | 0.35 | 0.58 | 0.46 | 0.36 | 0.47 | 0.11 |
| Fundamental A.E | -0.00 | -0.01 | 0.00 | -0.00 | 0.01 | -0.07 | -0.00 | 0.00 | 0.01 | 0.01 | 0.01 |
| Halo Effect | 0.35 | 0.22 | 0.35 | 0.31 | 0.07 | 0.11 | 0.12 | 0.09 | 0.14 | 0.12 | 0.03 |
| Hindsight Bias | 0.12 | 0.08 | 0.07 | 0.09 | 0.03 | 0.05 | 0.06 | 0.05 | 0.05 | 0.05 | 0.01 |
| Hyperbolic Discounting | -0.00 | 0.01 | -0.00 | 0.00 | 0.01 | 0.06 | 0.01 | -0.18 | -0.00 | -0.06 | 0.11 |
| Illusion of Control | 0.01 | 0.17 | 0.12 | 0.10 | 0.08 | -0.01 | 0.37 | 0.23 | 0.11 | 0.23 | 0.13 |
| In-Group Bias | 0.04 | 0.35 | 0.02 | 0.14 | 0.18 | 0.45 | 0.42 | 0.11 | 0.40 | 0.31 | 0.17 |
| Information Bias | 0.03 | -0.06 | 0.18 | 0.05 | 0.12 | 0.20 | -0.05 | -0.05 | -0.04 | -0.05 | 0.00 |
| Loss Aversion | 0.12 | 0.15 | 0.31 | 0.19 | 0.10 | -0.02 | -0.07 | -0.03 | -0.16 | -0.09 | 0.07 |
| Mental Accounting | -0.09 | -0.19 | -0.36 | -0.21 | 0.14 | -0.01 | -0.00 | 0.31 | 0.13 | 0.14 | 0.16 |
| Negativity Bias | -0.03 | -0.68 | 0.10 | -0.20 | 0.42 | 0.16 | 0.61 | 0.02 | 0.20 | 0.28 | 0.30 |
| Not Invented Here | 0.01 | -0.01 | -0.01 | -0.00 | 0.01 | -0.00 | 0.01 | 0.03 | -0.01 | 0.01 | 0.02 |
| Optimism Bias | -0.04 | 0.02 | 0.03 | 0.01 | 0.04 | -0.01 | 0.05 | -0.01 | 0.01 | 0.01 | 0.03 |
| Planning Fallacy | 0.09 | 0.09 | -0.16 | 0.00 | 0.14 | 0.07 | 0.16 | -0.02 | -0.02 | 0.04 | 0.11 |
| Reactance | -0.01 | -0.01 | -0.10 | -0.04 | 0.05 | -0.04 | 0.08 | -0.06 | -0.01 | 0.00 | 0.07 |
| Risk Compensation | 0.02 | 0.03 | -0.00 | 0.01 | 0.02 | -0.03 | -0.08 | -0.06 | -0.04 | -0.06 | 0.02 |
| Self-Serving Bias | 0.06 | 0.00 | 0.22 | 0.09 | 0.11 | 0.21 | 0.11 | 0.05 | 0.16 | 0.11 | 0.06 |
| Social Desirability Bias | -0.01 | -0.00 | 0.02 | 0.00 | 0.02 | -0.08 | 0.08 | 0.02 | 0.06 | 0.05 | 0.03 |
| Status-Quo Bias | -0.53 | -0.44 | 0.27 | -0.23 | 0.44 | -0.04 | 0.02 | 0.03 | -0.02 | 0.01 | 0.03 |
| Stereotyping | 0.02 | -0.01 | 0.00 | 0.00 | 0.02 | -0.10 | 0.01 | -0.00 | 0.04 | 0.02 | 0.03 |
| Survivorship Bias | 0.01 | 0.00 | 0.14 | 0.05 | 0.08 | 0.15 | 0.00 | 0.00 | 0.01 | 0.00 | 0.01 |
| Certainty | -0.15 | -0.02 | 0.16 | -0.00 | 0.16 | -0.13 | 0.00 | -0.06 | -0.15 | -0.07 | 0.08 |
| Belief Valid | -0.14 | -0.26 | -0.20 | -0.20 | 0.06 | -0.32 | -0.31 | -0.31 | -0.06 | -0.23 | 0.14 |
| MMLU | 0.45 | 0.46 | 0.45 | 0.46 | 0.00 | 0.47 | 0.47 | 0.47 | 0.47 | 0.47 | 0.01 |

Table 4: **Bias and Accuracy Scores Across Random Seeds for OLMo Models**: Bias scores and MMLU accuracy for OLMo-Tulu and OLMo-Flan models across three random seeds (columns 1–3). Each block includes the mean and standard deviation of the scores, along with the original score from the fully fine-tuned model OLMo-SFT ("Org"). The bias scores reflect the degree to which each model exhibits a specific behavioral bias, while the MMLU row represents task accuracy. This analysis highlights the variability introduced by random initialization.

## B  Verifying the Performance of Finetuned Models

Since we finetune the pretrained models also on their original instruction data, we can compare their performance to the original fine-tuned models. This comparison allows us to verify that our finetuning process, which involves compromises such as LoRa and downsampling of Flan, does not hinder their performance, ensuring they remain valid representatives of the fully fine-tuned models by keeping a reasonable MMLU score and having similarly directed bias scores.

| | T5-Flan | | | | | | T5-Tulu | | | | |
|---|---|---|---|---|---|---|---|---|---|---|---|
| **Bias** | 1 | 2 | 3 | Mean | Std | Org | 1 | 2 | 3 | Mean | Std |
| Anchoring | 0.16 | 0.10 | 0.19 | 0.15 | 0.05 | 0.33 | 0.28 | 0.29 | -0.01 | 0.18 | 0.17 |
| Anthropomorphism | -0.11 | 0.05 | 0.12 | 0.02 | 0.11 | -0.19 | 0.05 | -0.02 | 0.05 | 0.03 | 0.04 |
| Availability Heuristic | 0.05 | 0.05 | -0.03 | 0.02 | 0.04 | 0.03 | -0.05 | 0.02 | 0.00 | -0.01 | 0.04 |
| Bandwagon Effect | -0.05 | 0.05 | 0.03 | 0.01 | 0.05 | 0.25 | 0.02 | 0.10 | 0.31 | 0.15 | 0.15 |
| Confirmation Bias | -0.12 | -0.08 | 0.00 | -0.07 | 0.06 | 0.03 | 0.00 | -0.01 | 0.01 | 0.00 | 0.01 |
| Conservatism | 0.10 | 0.08 | -0.02 | 0.05 | 0.06 | -0.04 | -0.05 | 0.09 | 0.01 | 0.02 | 0.07 |
| Disposition Effect | -0.09 | 0.11 | -0.05 | -0.01 | 0.10 | -0.58 | -0.02 | 0.12 | 0.11 | 0.07 | 0.08 |
| Endowment Effect | 0.34 | -0.37 | -0.51 | -0.18 | 0.46 | -0.10 | -0.04 | -0.16 | 0.00 | -0.06 | 0.08 |
| Escalation of C. | -0.03 | 0.11 | 0.03 | 0.04 | 0.07 | 0.08 | 0.00 | 0.00 | 0.03 | 0.01 | 0.02 |
| Framing Effect | -0.06 | 0.09 | 0.29 | 0.11 | 0.18 | 0.44 | 0.14 | 0.22 | 0.04 | 0.13 | 0.09 |
| Fundamental A.E | -0.01 | 0.06 | 0.02 | 0.03 | 0.03 | -0.00 | -0.01 | 0.00 | -0.01 | -0.01 | 0.01 |
| Halo Effect | 0.05 | 0.10 | 0.18 | 0.11 | 0.07 | 0.23 | 0.07 | -0.02 | -0.03 | 0.00 | 0.06 |
| Hindsight Bias | -0.00 | 0.35 | 0.38 | 0.24 | 0.21 | 0.31 | 0.11 | 0.42 | 0.19 | 0.24 | 0.16 |
| Hyperbolic Discounting | 0.00 | -0.03 | 0.26 | 0.08 | 0.16 | 0.02 | -0.02 | -0.03 | 0.10 | 0.01 | 0.07 |
| Illusion of Control | -0.03 | 0.00 | 0.40 | 0.12 | 0.24 | 0.03 | 0.08 | -0.09 | 0.00 | -0.01 | 0.09 |
| In-Group Bias | -0.04 | -0.08 | 0.67 | 0.19 | 0.42 | 0.90 | 0.36 | 0.39 | 0.71 | 0.49 | 0.19 |
| Information Bias | 0.27 | 0.35 | 0.72 | 0.45 | 0.24 | 0.25 | -0.38 | 0.38 | 0.13 | 0.04 | 0.39 |
| Loss Aversion | 0.01 | -0.05 | -0.02 | -0.02 | 0.03 | -0.04 | 0.30 | 0.02 | 0.19 | 0.17 | 0.14 |
| Mental Accounting | 0.18 | -0.03 | -0.21 | -0.02 | 0.20 | 0.09 | 0.22 | 0.03 | -0.01 | 0.08 | 0.12 |
| Negativity Bias | 0.12 | 0.04 | 0.11 | 0.09 | 0.04 | -0.06 | -0.34 | 0.21 | -0.45 | -0.19 | 0.35 |
| Not Invented Here | -0.02 | -0.04 | -0.11 | -0.06 | 0.05 | 0.01 | 0.02 | -0.03 | -0.10 | -0.04 | 0.06 |
| Optimism Bias | -0.02 | 0.02 | 0.05 | 0.02 | 0.03 | -0.00 | 0.06 | 0.01 | -0.00 | 0.02 | 0.03 |
| Planning Fallacy | 0.20 | -0.09 | -0.01 | 0.03 | 0.15 | -0.06 | -0.00 | 0.08 | -0.03 | 0.01 | 0.06 |
| Reactance | 0.07 | 0.09 | 0.19 | 0.12 | 0.07 | -0.03 | -0.02 | 0.04 | 0.16 | 0.06 | 0.10 |
| Risk Compensation | -0.11 | 0.10 | 0.07 | 0.02 | 0.11 | 0.19 | 0.01 | -0.16 | -0.07 | -0.08 | 0.09 |
| Self-Serving Bias | 0.03 | 0.14 | 0.30 | 0.16 | 0.13 | 0.02 | 0.01 | 0.12 | 0.05 | 0.06 | 0.06 |
| Social Desirability Bias | -0.00 | -0.11 | 0.01 | -0.03 | 0.06 | -0.00 | 0.00 | 0.02 | 0.06 | 0.03 | 0.03 |
| Status-Quo Bias | -0.01 | 0.05 | 0.52 | 0.19 | 0.29 | 0.86 | 0.04 | -0.04 | -0.06 | -0.02 | 0.05 |
| Stereotyping | -0.23 | 0.02 | 0.03 | -0.06 | 0.15 | 0.13 | -0.00 | 0.01 | -0.04 | -0.01 | 0.03 |
| Survivorship Bias | 0.09 | 0.01 | -0.00 | 0.03 | 0.05 | 0.12 | 0.00 | -0.01 | 0.02 | 0.00 | 0.02 |
| Certainty | 0.25 | -0.01 | 0.11 | 0.12 | 0.13 | 0.17 | -0.07 | 0.12 | 0.16 | 0.07 | 0.12 |
| Belief Valid | 0.09 | 0.11 | 0.11 | 0.10 | 0.01 | 0.50 | 0.09 | 0.10 | 0.06 | 0.08 | 0.02 |
| MMLU | 0.51 | 0.51 | 0.51 | 0.51 | 0.00 | 0.55 | 0.48 | 0.46 | 0.47 | 0.47 | 0.01 |

Table 5: **Bias and Accuracy Scores Across Random Seeds for T5 Models**: Bias scores and MMLU accuracy for T5-Flan and T5-Tulu models across three random seeds (columns 1–3). Each block includes the mean and standard deviation of the scores, along with the original score from the fully fine-tuned model Flan-T5 ("Org"). The bias scores reflect the degree to which each model exhibits a specific behavioral bias, while the MMLU row represents task accuracy. This analysis highlights the variability introduced by random initialization.

Our OLMo-Tulu MMLU and bias scores have a similar direction as the original OLMo-SFT, and our T5–Flan also has similar scores to the original Flan–T5, as can be seen in Tables 4 and 5. The MMLU scores of our models close over 85% of the performance gap between the original base models (28.6 for OLMo, 23.0 for T5) and their fine-tuned counterparts. These results verify that our finetuning setting is good enough to create models that can simulate fully finetuned models, especially regarding the bias score trends.

## C  Finetuning Technical Details

Our finetuning and MMLU evaluation code is based entirely on the open-instruct code by AI2 (Ivison et al., 2024).[5] We conducted a small-scale hyperparameter search on a single seed for the finetuning processing, focusing on three key parameters: learning rate, LoRa rank, and LoRa $\alpha$ the scaling factor for LoRa weights upon merging. We applied LoRa weights on attention and feedforward layers to allow maximal flexibility for training. We explored ranks of 64, 128, 256, and 512, with the corresponding $\alpha$ set to the rank value or twice the rank, following standard practice. In parallel, we tested various learning rates ranging from $1 \times 10^{-6}$ to $1 \times 10^{-3}$, pruning the less successful configurations after 1000-2000 training steps using the MMLU score as the main criterion for effective hyperparameters.

For the remaining hyperparameters, we stick to the original finetuning settings of the OLMo-SFT and Flan-T5 models. Specifically, we maintained a context size of 2048 for OLMo, 1024 for instructions, and 128 for answers in T5. The batch sizes were 128 for OLMo and 64 for T5, with approximately 5,500 training steps for OLMo and 16,000 for T5, taking 5 and 10 days on 4 NVIDIA A40 GPUs. Additionally, we utilized fp16 precision for OLMo and bf16 for T5, with a LoRa dropout rate of 0.1 and a linear scheduling strategy that included a warm-up of 0.03.

| Base Model | Instruction Dataset | Source | Hugging Face Link |
|---|---|---|---|
| Llama2-7B | ShareGPT | AI2 | hf.co/AI2_LLAMA2_SHAREGPT |
| Llama2-7B | ShareGPT | Vicuna | hf.co/VICUNA_LLAMA2_SHAREGPT |
| Llama2-7B | Tulu-2 | AI2 | hf.co/AI2_LLAMA2_TULU |
| Mistral-7B | ShareGPT | DICE | hf.co/DICE_MISTRAL_SHAREGPT |
| Mistral-7B | Tulu-2 (v1) | DICE | hf.co/DICE_MISTRAL_TULU1 |
| Mistral-7B | Tulu-2 (v2) | DICE | hf.co/DICE_MISTRAL_TULU2 |

Table 6: **Community-Finetuned External Models Used for Validation**: Six fully finetuned models built on Llama2-7B and Mistral-7B. Each model was finetuned on either Tulu-2 or ShareGPT by independent groups. These models were not selected for bias separation or trained using LoRA, offering a realistic validation of our findings.

### C.1  External Models

We include additional fully finetuned models to test the generalizability of our findings beyond the controlled LoRA setup. These models were released by different community groups and trained with varying instruction datasets and recipes, providing a natural source of external validation. Table 6 lists the six models used.

## D  Analysis Details

This section presents the complete results tables, including individual bias scores for each of the 12 fine-tuned models, as well as the original OLMo-SFT and Flan-T5 models. Table 4 reports scores for OLMo models, and Table 5 for T5 models. Table 7 provides aggregated results across random seeds, highlighting their alignment with the fully fine-tuned models.

---

[5]https://github.com/allenai/open-instruct

| Bias | OLMo-Tulu | | | | | T5-Flan | | | | |
|---|---|---|---|---|---|---|---|---|---|---|
| | Full-FT | Mean | Median | Majority | Agg | Full-FT | Mean | Median | Majority | Agg |
| Anchoring | 0.06 | 0.04 | 0.02 | True | True | 0.33 | 0.15 | 0.16 | True | True |
| Anthropomorphism | -0.07 | -0.05 | -0.04 | True | True | -0.19 | 0.02 | 0.05 | False | False |
| Availability H. | 0.14 | 0.19 | 0.18 | True | True | 0.03 | 0.02 | 0.05 | True | True |
| Bandwagon Effect | 0.02 | 0.09 | 0.02 | True | True | 0.25 | 0.01 | 0.03 | False | False |
| Confirmation Bias | -0.02 | 0.04 | 0.03 | True | True | 0.03 | -0.07 | -0.08 | True | True |
| Conservatism | -0.01 | -0.01 | 0.00 | True | True | -0.04 | 0.05 | 0.08 | True | True |
| Disposition Effect | 0.05 | 0.05 | -0.03 | True | True | -0.58 | -0.01 | -0.05 | False | False |
| Endowment Effect | -0.01 | -0.55 | -0.49 | False | False | -0.10 | -0.18 | -0.37 | True | True |
| Escalation of C. | -0.01 | 0.06 | 0.05 | True | True | 0.08 | 0.04 | 0.03 | True | True |
| Framing Effect | 0.35 | 0.52 | 0.60 | True | True | 0.44 | 0.11 | 0.09 | True | True |
| Fundamental A.E | -0.07 | -0.00 | -0.00 | True | True | -0.00 | 0.03 | 0.02 | True | True |
| Halo Effect | 0.11 | 0.31 | 0.35 | True | True | 0.23 | 0.11 | 0.10 | True | True |
| Hindsight Bias | 0.05 | 0.09 | 0.08 | True | True | 0.31 | 0.24 | 0.35 | True | True |
| Hyperbolic Discounting | 0.06 | 0.00 | -0.00 | True | True | 0.02 | 0.08 | 0.00 | True | True |
| Illusion of Control | -0.01 | 0.10 | 0.12 | False | False | 0.03 | 0.12 | 0.00 | True | True |
| In-Group Bias | 0.45 | 0.14 | 0.04 | False | False | 0.90 | 0.19 | -0.04 | False | False |
| Information Bias | 0.20 | 0.05 | 0.03 | False | False | 0.25 | 0.45 | 0.35 | True | True |
| Loss Aversion | -0.02 | 0.19 | 0.15 | False | False | -0.04 | -0.02 | -0.02 | True | True |
| Mental Accounting | -0.01 | -0.21 | -0.19 | False | False | 0.09 | -0.02 | -0.03 | False | False |
| Negativity Bias | 0.16 | -0.20 | -0.03 | False | False | -0.06 | 0.09 | 0.11 | False | False |
| Not Invented Here | -0.00 | -0.00 | -0.01 | True | True | 0.01 | -0.06 | -0.04 | True | True |
| Optimism Bias | -0.01 | 0.01 | 0.02 | True | True | -0.00 | 0.02 | 0.02 | True | True |
| Planning Fallacy | 0.07 | 0.00 | 0.09 | False | True | -0.06 | 0.03 | -0.01 | False | True |
| Reactance | -0.04 | -0.04 | -0.01 | True | True | -0.03 | 0.12 | 0.09 | False | False |
| Risk Compensation | -0.03 | 0.01 | 0.02 | True | True | 0.19 | 0.02 | 0.07 | False | False |
| Self-Serving | 0.21 | 0.09 | 0.06 | False | False | 0.02 | 0.16 | 0.14 | False | False |
| Social Desirability | -0.08 | 0.00 | -0.00 | True | True | -0.00 | -0.03 | -0.00 | True | True |
| Status-Quo Bias | -0.04 | -0.23 | -0.44 | False | False | 0.86 | 0.19 | 0.05 | False | False |
| Stereotyping | -0.10 | 0.00 | 0.00 | False | False | 0.13 | -0.06 | 0.02 | False | False |
| Survivorship | 0.15 | 0.05 | 0.01 | False | False | 0.12 | 0.03 | 0.01 | False | True |
| Certainty | -0.13 | -0.00 | -0.02 | False | False | 0.17 | 0.12 | 0.11 | True | True |
| Belief Valid | -0.32 | -0.20 | -0.20 | True | True | 0.50 | 0.10 | 0.11 | True | True |
| Avg Diff | - | 0.12 | 0.13 | 63.33% | 66.67% | - | 0.18 | 0.19 | 60.00% | 66.67% |

Table 7: **Comparison of Aggregated Bias Scores**: How bias scores change when aggregating results across different random seeds. The table compares the original bias score with the aggregated scores using mean, median, majority vote agreement, and aggregate similarity. A majority vote agreement means the bias direction matches the original if most seeds agree, while aggregate similarity considers a match if the majority vote is True or if the mean or median score is within a set threshold of the original. The last row summarizes the average difference between the original score and the mean and median, along with the agreement percentages for majority vote agreement and aggregate similarity.

| | | Cluster Quality | | | | |
|---|---|---|---|---|---|---|
| Granularity | Clustering | Silhouette (↑) | Calinski. (↑) | Davies. (↓) | Intra D. (↓) | Inter D. (↑) |
| | Random | -0.001 | 1.017 | 2.043 | 32.022 | 31.996 |
| Scenario | Instruction | 0.014 | 1.056 | 1.937 | 31.775 | 32.161 |
| Level | Pretraining | **0.096** | **1.660** | **1.547** | **30.074** | **33.295** |
| | K-Means | 0.089 | 1.543 | 1.453 | 30.457 | 33.363 |

Table 8: Clustering quality metrics for community-finetuned models (Llama2/Mistral) using scenario-level bias vectors. **Pretraining-based clustering again outperforms instruction and random baselines, replicating the trend observed in the controlled setup.** K-means provides a strong unsupervised reference.

## D.1 Distinct Bias Scores

To determine whether two bias scores differ meaningfully, we must account for statistical noise. One such case involves computing the majority vote on bias direction across random seeds by categorizing scores as positive, negative, or neutral. Since bias scores range from $[-1, 1]$, we define a neutral zone to filter out near-zero values that may reflect noise rather than consistent trends.

To define this threshold, we conduct an independent two-sample t-test, assessing whether the difference in means between two groups is statistically significant. With each group containing $n = 1000$[6] values, the standard error (SE) of the mean difference is:

$$SE = \sqrt{\frac{2\sigma^2}{n}}.$$ (6)

Assuming the maximum standard deviation $\sigma = 1$, this simplifies to:

$$SE \approx 0.045.$$ (7)

For significance at $p < 0.05$, the t-statistic must satisfy:

$$|t| = \frac{|Score|}{SE} > 1.96.$$ (8)

Substituting $SE \approx 0.045$, we find:

$$|Score| > 0.088.$$ (9)

Thus, bias scores within $[-0.088, 0.088]$ are considered statistically indistinguishable from zero and classified as neutral. Likewise, two scores are considered equivalent if their absolute difference is below this threshold.

## D.2 Clustering Analysis

This section presents additional results from the clustering analysis. Figure 7 shows the mean bias score per cluster for each pretrained model, providing a visual representation of the bias patterns among models sharing the same pretraining. Figure 6 shows a 2D PCA projection of the bias vectors, labeled using K-Means clustering based on bias-level scores.

---

[6]Certainty and belief bias scores are adjusted based on their respective sample sizes.

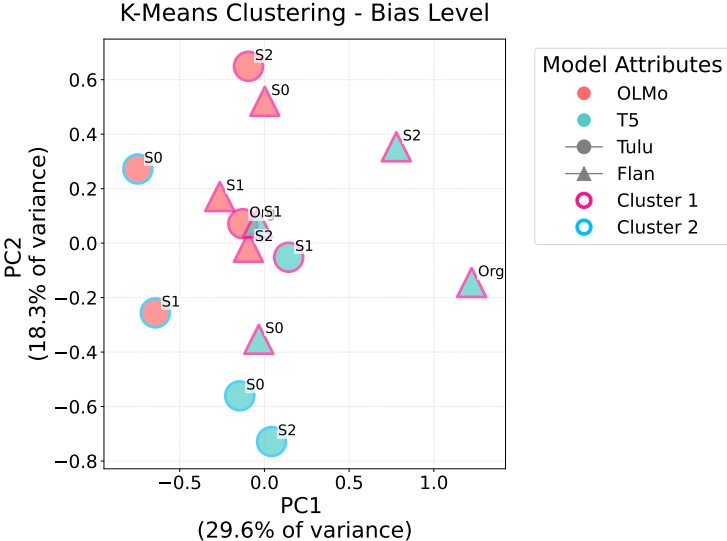

Figure 6: **Clustering PCA analysis of model bias vectors at the bias level**. Each point represents a model, where the fill color indicates the pretrained model (OLMo in pink, T5 in turquoise), the shape shows the instruction tuning type (Tulu - circle, Flan - triangle), and the edge color represents the K-Means cluster assignment. Labels indicate model identifiers, where 'Org' represents the original model and S0–S2 denote different seeds. The plot shows that the diagonal formed by PC1 and PC2 effectively separates models by their pretraining origin, indicating that base model characteristics persist after instruction tuning. PC1 explains 29.6% of variance, and PC2 explains 18.3% , capturing much of the variation linked to pretraining differences.

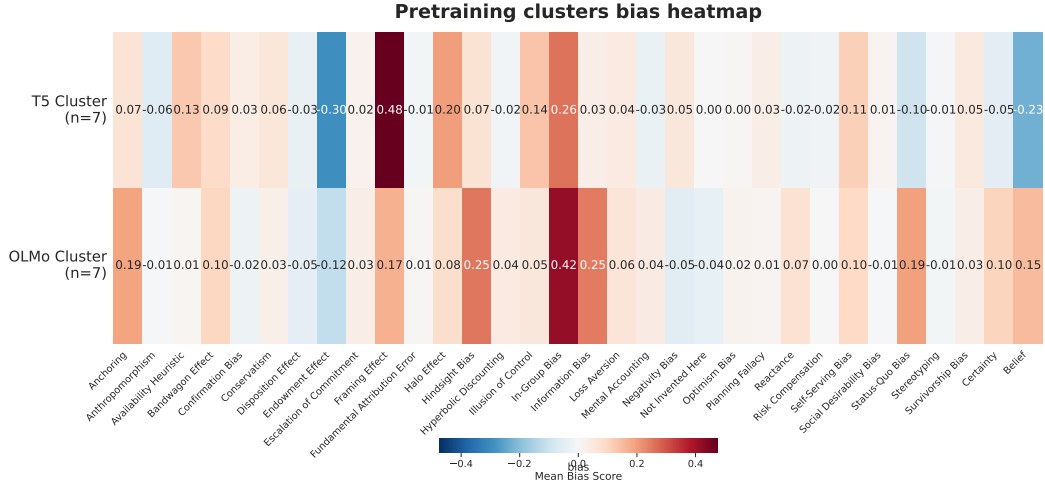

Figure 7: **Mean bias score per cluster grouped by pretraining model**.

**External models results.** We report the full clustering metrics for the six community-finetuned models described in Appendix C.1. As shown in Table 8, clustering by pretraining identity again outperforms clustering by instruction dataset across all evaluation metrics. These results mirror those observed in our controlled setup (Table 2), reinforcing the conclusion that pretraining plays a dominant role in shaping cognitive bias patterns. This replication, despite differences in model architecture, training recipes, and finetuning methods, provides strong external validation of our findings.

