# OpenReview forum: "Planted in Pretraining, Swayed by Finetuning: A Case Study on the Origins of Cognitive Biases in LLMs"
_colmweb.org/COLM/2025/Conference — COLM 2025_

### Official Review · Reviewer_mav9 · 2025-05-12

**Rating:** 7
**Confidence:** 4
**Ethics Flag:** 1

**Summary:**

The paper carries out a causal analysis to attempt to determine whether cognitive-bias-like behavior in LLMs originates in pretraining data, finetuning data, or randomness that occurs during training.

**Reasons To Accept:**

* The paper addresses an important question, which gives it the potential to be impactful.
* The approach taken to address the question is sound and carefully thought-out.
* The study uses a causal analysis framework, which is rare in the field.
* The study carries out statistical tests, which is rare in the field.
* The paper is clear and honest about the limitations of the approach even outside the limitations section.
* The paper is mostly clear and well-written.

**Reasons To Reject:**

* I don't think the paper successfully argues for training randomness influencing observed bias but not latent bias. Based on the paper, it is unclear how the two would be disentangled, or even what it would mean for there to be an influence on observed but not latent bias. In principle, I could imagine an argument that the randomness in the training could interact with idiosyncratic features of the test set which would give the appearance of the bias, but this doesn't seem to be the argument that is being made. Additionally, the paper argues that this randomness is "unlikely to introduce consistent or substantive internal changes", but the large standard deviations of scores across seeds reported for some of the biases in the Appendix seem to suggest that the models can show quite large (and thus at least to some extent relatively consistent) biases that vary based only on random seed.
* The other main concern is the assumption that the differences in bias between OLMo and T5 can be attributed to differences in pretraining data. While there is no doubt that it plays a role, the two models differ in other ways, most notably architecture: OLMo is an autoregressive transformer and T5 is a seq2seq model. They also differ in size (parameters). Each of these may interact with pretraining data in idiosyncratic ways, especially given that the pretraining datasets of each language model are drastically different in size: T5 is trained on 34B tokens, while the original OLMo was trained on about 2.5T tokens. Altogether, it's hard to determine whether the approach taken in the paper allows for the straightforward apples-to-apples comparison suggested.
* Relatedly, the paper is limited in that the authors do not provide any clear theories, hypotheses, or predictions about how training data should impact cognitive biases in language models. While it may be unreasonable to expect the authors to have a complete theory of how specific features of training data may lead to different degrees of each individual bias, the paper could benefit from at least some consideration of this.
* The Discussion and Conclusion are limited - there is insufficient discussion about the implications of the work.
* The authors do not provide any of their research materials.

---

> ### Author Response · Authors · 2025-05-30
>
> Thank you for your thoughtful and constructive review. We appreciate your recognition of the paper’s contributions, including the importance of the research question, the soundness of the causal analysis, and the clarity of presentation. Your comments have been very helpful in identifying areas for clarification and improvement.
>
>
> - **On randomness and latent vs. observed bias** -
> We intended latent bias to refer to meaningful, underlying patterns learned during pretraining, while observed bias reflects how these patterns are expressed in outputs. Our assumption was that training randomness influences the latter without substantially altering the former. Our assumption was motivated by the idea that randomness in weight initialization, data order, and optimizer behavior typically introduces noise around stable learned representations, rather than reshaping the core behavioral patterns acquired through pretraining.
> That said, as you noted, variability across seeds could potentially reflect more directional-seeming influences. However, our analysis in Step 1 (Section 5.1) directly addresses this by quantifying the variance across runs. We show that, despite some fluctuation, the overall bias patterns remain largely consistent. We will revise the paper to clarify this assumption and analysis, acknowledge its limitations, and discuss how randomness may sometimes interact with model behavior more systematically.
>
> - **On model differences beyond pretraining** -
> We agree that architectural and size differences between OLMo and T5 complicate attributing bias differences solely to pretraining. To address this, as noted in the general response, we added experiments using Mistral-7B and Llama-2-7B models, which are more comparable. These replicate our main findings, and we will include them in the revision to strengthen our conclusions.
>
> - **Lack of theoretical framing** -
> Thank you for highlighting this. These are important and fascinating directions that we are excited to reflect on more explicitly. While it may be beyond the scope of this work to develop a full theory connecting specific data properties to individual biases, we agree that the paper would benefit from some discussion of these possibilities. In the revised version, we plan to reflect on potential contributing factors. For example, we plan to include the following statement in the discussion: “While our work does not model specific causal links between training data properties and individual biases, several factors such as corpus composition, linguistic framing, tokenization choices, filtering decisions, and sampling strategies may plausibly influence the emergence of cognitive biases during pretraining (Feng et al., 2023; Silva et al., 2021; Navigli et al., 2023; Chang et al., 2024). Exploring these connections is a promising direction for future work.” Such additions will help frame promising avenues for future work.
>
> - **Lack of discussion on implications and mitigation strategies** -
> Concerning the implications of our work, there are several promising mitigation directions worth discussing. Since our results indicate that pretraining is the primary source of bias, early-stage interventions may be especially effective. For example, instance attribution methods could help identify and filter out harmful training examples, cognitive bias scores could be monitored during pretraining to enable early intervention, and small-scale pretraining runs on alternative datasets could guide better data selection.
> At the same time, instruction-phase interventions remain highly relevant, as they are more practical to apply and can still meaningfully influence how pretraining biases are expressed. Techniques such as incorporating self-helping prompts during instruction tuning (Echterhoff et al., 2024), like how chain-of-thought prompting aids reasoning, could help reduce surfaced biases. Another avenue involves curating instruction datasets to promote internal consistency and minimize exposure to contradictory or biased patterns. Additionally, our findings raise the possibility that instruction tuning alone may be insufficient, and that stronger post-training alignment methods, such as variants of RLHF that reward unbiased rather than merely preferred outputs, may be necessary. We will revise the concluding section to incorporate these directions.
>
>
> Thank you again for your valuable feedback. Your comments have directly improved the clarity and scope of our work, and we hope this response addresses some of the concerns.

---

> > ### Author Response · Authors · 2025-05-30
> >
> > - **On the availability of research materials** -
> > We apologize for the oversight. All code, data, model weights, predictions, metadata, and analysis scripts will be publicly released to ensure full reproducibility and transparency. We are excited about the possibility that these resources could be useful to the broader research community.
> > Aside from analysis, much of our implementation builds on existing open-source efforts. Specifically, our finetuning and MMLU evaluation pipeline is based on the Open-Instruct codebase by AI2 (Ivison et al., 2024):
> > (https://github.com/allenai/open-instruct )
> > Our bias evaluation code builds on the original repositories associated with the benchmark datasets:
> > (https://github.com/simonmalberg/cognitive-biases-in-llms ; https://github.com/itay1itzhak/InstructedToBias )

---

> > ### Comment · Reviewer_mav9 · 2025-06-07
> >
> > Thank you to the authors for their detailed response to the concerns raised in my initial review. The additional studies clearly support the other results and demonstrate that the generalizability of the findings.
> >
> > However, I am still not sure that I find the theoretical elements of the paper convincing. In particular, I still do not think that the distinction between latent and observed bias is clear in a way that is empirically testable, theoretically helpful, and non-tautological. Specifically, even if the effect of randomness on model behavior is not predictable, I don't think it follows (or is demonstrated by the study) that it is not consistent within a given model, and thus, it is still not clear to me that the observed and latent biases are distinguishable.
> >
> > Another note is that while the study convincingly shows that the original pretrained model has a strong and systematic impact on the 'cognitive biases' displayed by the finetuned model, referring to this as the effect 'pretraining' more generally seems like a logical leap to me without a more clear definition of what factors are encompassed by 'pretraining' (e.g., the factors mentioned in the 'Lack of theoretical framing' response above). I think that it would benefit the paper to be clear about this early on, as the empirical results are nonetheless interesting.
> >
> > Thus, in general, I still have concerns about the theoretical elements of the paper, and I think the implications and the extent to which the results should be expected to generalize still require additional discussion.
> >
> >
> > **Nonetheless, I think the empirical findings of the paper are strong and valuable** - the experiments are well-designed, the analyses are sound, and the main results are clear. I believe that it would be valuable for the paper to be presented, and I think that it would be of interest to the community. For this reason, I am raising my overall score. However, I would ask the authors to further consider the framing and discussion of the results.

---

> > ### Author Response · Authors · 2025-06-08
> >
> > Thank you for your thoughtful follow-up and for your willingness to engage deeply with both the empirical and theoretical aspects of our work. This discussion has been intriguing and surfaced theoretical considerations that have been underemphasized in the original submission. We also greatly appreciate your updated score and your recognition of the empirical contributions and potential impact of the paper.
> >
> > - **On the latent vs. observed bias distinction** -
> > We fully understand your continued concerns about this distinction. Our intention was not to propose a rigid or directly observable binary, but rather to offer a conceptual framing to reason about where bias arises and how it manifests. We recognize that for this framing to be useful, it must not only align with experimental design but also offer theoretical clarity. In the revision, we will explicitly acknowledge the limitations of treating latent and observed bias as cleanly separable and clarify the assumptions behind this framing. We will make sure to avoid implying that the distinction is theoretically clean or empirically resolvable in a strict sense.
> >
> > - **On the use of "pretraining" as a causal factor** -
> > We understand your concern that referring to "pretraining" as a monolithic cause without specifying its components can be misleading. We will revise the beginning of the paper and the discussion to clarify that our findings reflect differences between the pretrained model checkpoints used, not a controlled manipulation of specific aspects of the whole pretraining process. We will also integrate a short reflection, as mentioned in our earlier response, pointing to possible contributing factors such as corpus composition, tokenization, filtering, and sampling strategies. We will emphasize that isolating their individual effects remains an important direction for future work.
> >
> > Thank you again for your careful reading and constructive feedback. We appreciate your suggestion to refine the theoretical framing further and clarify the scope of our claims. We believe these changes will meaningfully improve the clarity and impact of the paper, and we are grateful for your thoughtful engagement throughout the review process.

---

### Official Review · Reviewer_Eywr · 2025-05-12

**Rating:** 7
**Confidence:** 2
**Ethics Flag:** 1

**Summary:**

This paper investigates the origins of cognitive biases in LLMs, focusing on disentangling the contributions of pretraining, finetuning data, and training randomness. The authors propose a two-step causal experimental approach: first, they finetune models multiple times using different random seeds to study the impact of training randomness on over 30 cognitive biases; second, they introduce cross-tuning, where they swap instruction datasets between models to isolate bias sources. The experiments are conducted using OLMo-7B and T5-11B, and the results indicate that pretraining is the primary cause of cognitive biases, with training randomness introducing only minor variability. The paper provides valuable insights into understanding and mitigating biases in LLMs.

**Questions To Authors:**

NA

**Reasons To Accept:**

- The findings have important implications for developing strategies to mitigate cognitive biases in LLMs.
- The two-step causal experimental approach is innovative and well-structured. The use of cross-tuning to isolate the effects of pretraining and finetuning data is particularly clever and provides a robust methodology for investigating the origins of cognitive biases.
- The paper is well-written and easy to read. The results are presented clearly and convincingly. The clustering analysis effectively demonstrates that pretraining has a more substantial impact on bias patterns than finetuning data, which is a significant contribution to the field.

**Reasons To Reject:**

- The study is limited to two base models (OLMo-7B and T5-11B) with different architectures and two instruction finetuning datasets. While these models provide valuable insights, the results may not generalize to other models or datasets. Although the authors acknowledge this in the Limitations section, I believe this remains a notable constraint on the broader applicability of the results.

---

> ### Author Response · Authors · 2025-05-30
>
> Thank you for the thoughtful and encouraging review. We're especially grateful for your recognition of our causal framework, the clarity of our writing, and the significance of our findings.
>
> We fully agree that generalizability is critical. As detailed in our general comment, to address this limitation, we added experiments on additional models based on Mistral and Llama2, finetuned on Tulu 2 and ShareGPT. Despite being trained in uncontrolled settings, these models **replicate our core finding: pretraining is the primary driver of bias**. We hope this stronger empirical support encourages you to consider raising your score.
>
> Thank you again for your constructive feedback. This highlight has helped us improve both the depth of the paper, and we’d be glad to hear if they address your concerns.

---

> > ### Comment · Reviewer_Eywr · 2025-06-10
> >
> > Thank you for your response. I hope those additional experiments presented in the general response will be included in the final version of the paper.

---

> > > ### Author Response · Authors · 2025-06-10
> > >
> > > Thank you for your follow-up. We're glad the additional experiments were helpful, and we confirm that they will be included in the final version of the paper.
> > >
> > > If you feel they strengthen the work, would you consider updating your score to reflect that?

---

> ### Comment · Area_Chair_uhNd · 2025-06-08
>
> Hi reviewer! A quick reminder that the discussion period ends on 6/10. Please make sure to read and respond to the authors' rebuttals -- It'd mean a lot to the authors who have put in a lot of efforts into their work!

---

### Official Review · Reviewer_VEAg · 2025-05-13

**Rating:** 7
**Confidence:** 4
**Ethics Flag:** 1

**Summary:**

The paper defines a controlled 2-step causal framework to isolate origins of cognitive biases in 2 publicly available LLMs (training randomness using seeds, cross-tuning to decouple pre-training vs instruction tuning).The paper defines most of the important terms involved in the discussion, with supporting equations and intuitive figures.

Cross-tuning (swapping instruction datasets between models), defining a distinction between bias emergence vs bias surfacing, use of clustering metrics to evaluate bias consistency.

The work challenges the idea that instruction tuning is a major cause of bias in LLMs, and instead focuses on the pertaining stage. This could be useful for current work on data selection and bias aware training objectives.

**Reasons To Accept:**

I like how the study is structured to decompose experiments to understand the impacts of random seeds, instruction tuning data, and pretraining, with reasonable baselines like K-Means, and experimental rigor in terms of statistical validity.

**Reasons To Reject:**

While OLMo and T5 make sense to use for the research questions posed in the paper, I am wondering if we would see similar conclusions from other widely used set of models like Llama.

While the paper clusters bias vectors together, it does not look into the different types of cognitive biases and how these finger-grained scales would vary with pre-training vs fine-tuning. Knowing these might help making the mitigation strategies more actionable, by clarifying which specific biases show up at which phase of training. Similarly aggregating bias scores across multiple scenarios might mask variance across different contexts and I would have liked to see more analysis to pinpoint specific scenarios which show more pretraining bias.

The cross-tuning analysis assumes that the instruction tuning datasets are meaninfully different without qualitative comparisons of things like domain coverage, or words related to bias. If the datasets are similar, then maybe we cannot say for sure if the observed persistence of bias is actually due to pre-training bias domination or simply because there is insufficient difference between fine-tuning datasets.

---

> ### Author Response · Authors · 2025-05-30
>
> Thank you for the detailed and thoughtful review. We're glad you appreciated the structure of the study, the clarity of definitions and visualizations, and the experimental rigor.
>
> - **On model and dataset scope** -
> As detailed in our general comment, we added experiments on six additional models based on Mistral and Llama2, finetuned on Tulu 2 and ShareGPT. Despite being trained in uncontrolled settings, these models **replicate our core finding: pretraining is the primary driver of bias**. We hope this stronger empirical support encourages you to consider raising your score.
>
> - **On bias type granularity and scenario variance** -
> We agree this is an important and interesting direction. A complete table of per-bias results is included in Appendix D, allowing such fine-grained analysis. Our preliminary analysis found that most individual biases follow the overall clustering pattern, with no consistent deviations. One minor exception was Status-Quo Bias (a tendency to prefer the current state over change), where Flan-models (0.01, 0.19) showed slightly stronger bias than Tulu-models (-0.02, -0.23), though this difference had high variance. Regarding scenario-level analysis, including full results is infeasible due to the large number of scenarios and the difficulty of drawing clear patterns when considering each bias individually. We will clarify this in the analysis section.
> We will include this individual biases analysis in the revised paper and expand the discussion to note that some bias types may be more or less sensitive to pretraining vs. finetuning. We will also clarify that while we aggregate across biases, this may mask variation, and cases like Status-Quo Bias highlight where divergence can still emerge. Thank you for raising this point.
>
> - **On instruction dataset similarity** -
> You raise an important point. While the instruction datasets (Flan, Tulu2) are often treated as distinct, we did not include a formal content analysis. In our case, the instruction datasets were Flan and Tulu 2. Tulu 2 was explicitly constructed to improve upon and deviate from Flan, aiming to reduce known limitations and increase logical performance. We will clarify this motivation in the paper and include a brief comparison of dataset characteristics to better support the cross-tuning setup. Additionally, as described in the general comment, we expanded our experiments to include models finetuned on ShareGPT, a dataset derived entirely from real user interactions. This further increases the diversity of instruction data and strengthens the validity of our conclusions.
>
>
> Thank you again for your insightful feedback. These points have helped us improve both the clarity and depth of the paper. We’d be glad to hear if they address your concerns.

---

> > ### Comment · Reviewer_VEAg · 2025-06-09
> >
> > Thank you for your response. I acknowledge that the authors have addressed my concerns.

---

> > ### Author Response · Authors · 2025-06-10
> >
> > Thank you for the follow-up. We're glad to hear our response addressed your concerns. All changes will be included in the final version of the paper.
> >
> > Would you consider updating your score to reflect the improvements?

---

### Official Review · Reviewer_BmJn · 2025-05-16

**Rating:** 8
**Confidence:** 4
**Ethics Flag:** 1

**Summary:**

The paper presents a causal framework to assess how pretraining, finetuning data, and training randomness influence cognitive biases in language models. Using systematic experiments with OLMo-7B and T5-11B, the authors highlight pretraining as the primary contributor to these biases.

**Questions To Authors:**

N/A

**Reasons To Accept:**

Below are the following reason for acceptance:
- The paper presents a novel causal framework that effectively isolates and identifies distinct sources of bias in large language models, marking a significant methodological improvement over prior approaches.
- This work fills a crucial gap in our understanding of bias in LLMs, offering valuable insights for designing stronger fairness interventions and building more trustworthy AI systems.
- The author exhibit clarity in methodology, visualization, and insightful analysis.

**Reasons To Reject:**

- The study’s reliance on a narrow range of models and datasets may limit the broader applicability of its findings, potentially reducing their relevance across diverse architectures or domains.
- While the paper offers a strong diagnostic framework, it provides minimal discussion on actionable mitigation strategies, leaving a gap in translating insights into practical interventions.

---

> ### Author Response · Authors · 2025-05-30
>
> Thank you for the thoughtful review and for highlighting the clarity of our methodology, the insightful analysis, and the importance of our framework for understanding bias in LLMs.
>
> - **On model and dataset scope** -
> As detailed in our general comment, we added experiments on additional models based on Mistral and Llama2, finetuned on Tulu 2 and ShareGPT. Despite being trained in uncontrolled settings, these models **replicate our core finding: pretraining is the primary driver of bias**. We hope this stronger empirical support encourages you to consider raising your score.
>
> - **On mitigation strategies** -
> While our primary goal was to isolate the source of bias, we agree that discussing how these insights can inform mitigation is important. We will expand the discussion to suggest concrete directions.
> Given that our findings point to pretraining as the primary source of bias, mitigation efforts at this stage may be especially impactful. For example, instance attribution techniques could help identify and remove training examples that disproportionately contribute to bias. Monitoring cognitive bias scores during pretraining could enable early intervention by stopping, reverting, or adjusting training when problematic trends are detected. Additionally, small-scale pretraining experiments on different datasets or data compositions could help identify sources that lead to reduced bias.
> Instruction-phase interventions, while operating later in the pipeline, offer greater flexibility and lower resource demands, making them attractive for practical deployment. For example, following the success of self-helping prompts in reducing bias (Echterhoff et al., 2024), one promising approach is to integrate such prompts during instruction tuning, similar to how chain-of-thought prompting improves reasoning. Another direction is to curate instruction data to encourage logically consistent, unbiased behavior and reduce contradictory examples. While exploratory, we believe these ideas can help translate diagnostic insights into actionable interventions.
>
> Thank you again for your constructive suggestions. We believe these additions significantly strengthen the paper, and we’d be happy to hear if they resolve your concerns.

---

> > ### Comment · Reviewer_BmJn · 2025-06-08
> >
> > Thank you for the careful clarifications and for running the additional experiments on Mistral- and Llama2-based checkpoints. Seeing the same causal pattern emerge across four architectures (OLMo-7B, T5-11B, Mistral, Llama2) goes a long way toward allaying my concern that the original findings might have been model or dataset-specific. Please be sure the camera-ready clearly marks these new results and describes any caveats (e.g., instruction-tuning details).
> >
> > The empirical results presented are robust, with well-designed experiments and thorough analyses. The key findings are clearly articulated and supported by the evidence. I believe this work merits presentation and will be of significant interest to the community. Accordingly, I have revised my overall score upward.

---

> > > ### Author Response · Authors · 2025-06-10
> > >
> > > Thank you for your thoughtful follow-up and for recognizing the value of the additional experiments. We’ll be sure to clearly highlight the new results and relevant details in the camera-ready. We truly appreciate your acknowledgment and are grateful for the updated score.

---

### Comment · Program_Chairs · 2025-04-03

This paper violates the page limit due to adding a limitation sections beyond the page limit. COLM does not have a special provision to allow for an additional page for the limitations section. However, due to this misunderstanding being widespread, the PCs decided to show leniency this year only. Reviewers and ACs are asked to ignore any limitation section content that is beyond the 9 page limit. Authors cannot refer reviewers to this content during the discussion period, and they are not to expect this content to be read.

---

### Author Response · Authors · 2025-05-30

We sincerely thank all reviewers for their thoughtful and constructive feedback. A common concern raised relates to the generalizability of our findings, given that the original study included only two base models (OLMo-7B and T5-11B) with different architectures and two instruction datasets.

To directly address this important concern, we conducted additional experiments using publicly released fully finetuned models, built from two new pretrained base models: **Mistral-7B** (Jiang et al., 2023) and **Llama-2-7B** (Touvron et al., 2023). Each was finetuned on two instruction datasets: Tulu 2 and ShareGPT (Chiang et al., 2023), with two combinations having two different variants (Llama2-ShareGPT and Mistral-Tulu), amounting to a total of six models. These models were not screened in advance for bias separability, unlike our original study. Additionally, they were not finetuned under our controlled setup, nor via LoRa, which introduces variability and enhances external validity.

We observed the **same pattern of results in the additional experiments**:

- Clustering by pretraining identity again outperformed clustering by instruction dataset and random seed (see table below).
- K-means clustering is closely aligned with the pretraining basis.
- PCA again showed that the first principal component captured pretraining-specific bias variation (see figure link below).

These findings replicate the main result of our paper. Even in a natural setting with community-trained models, **cognitive bias patterns are more strongly shaped by pretraining than finetuning**.

We will incorporate these results into the revised version of the paper, along with figures and discussions. We believe these additions strengthen the case for the robustness and broader applicability of our conclusions. We hope this extension encourages reviewers to reconsider their evaluations in light of the expanded empirical support. We thank the reviewers once again for helping us improve the clarity, rigor, and scope of this work.

Table:

Metric (better)                       | Unsupervised | Pretraining | Instruction | Random Seed
--------------------------------------|--------------------|----------------|---------------|--------------
Silhouette Score (↑)              | 0.089              | **0.096**   | 0.014         | -0.001
Calinski-Harabasz Score (↑) | 1.543              | **1.660**   | 1.056         | 1.017
Davies-Bouldin Score (↓)      | 1.453              | **1.547**   | 1.937         | 2.043
Intra-cluster Dist (↓)               | 30.457            | **30.074** | 31.775      | 32.022
Inter-cluster Dist (↑)               | 33.363            | **33.295** | 32.161      | 31.996


PCA Figure:
https://imgur.com/a/cxlSHPk

Models links
Mistral-Tulu:
https://huggingface.co/nyu-dice-lab/Mistral-7B-Base-SFT-Tulu2 , https://huggingface.co/nyu-dice-lab/Mistral-7B-Base-SFT-Tulu2-2.0
Mistral-ShareGPT: https://huggingface.co/nyu-dice-lab/Mistral-7B-Base-SFT-ShareGPT-Vicuna
Llama2-Tulu:
https://huggingface.co/allenai/tulu-2-7b
Llama-ShareGPT:
https://huggingface.co/allenai/open-instruct-llama2-sharegpt-7b
https://huggingface.co/lmsys/vicuna-7b-v1.5

---

### Author Response · Authors · 2025-06-06
**Authors Response**

Thank you again for your helpful feedback. We've done our best to address your comments and would be glad to hear your thoughts or discuss further if anything remains unclear.

---

### Decision · Program_Chairs · 2025-07-08

**Decision:**

Accept

**Comment:**

This paper uses a cross-tuning causal probe to show that most cognitive biases observed in LLMs come from the pre-training stage rather than from later instruction-tuning. All reviewers liked the paper and most concerns are successfully addressed through rebuttal:

Strengths:
1. Novel causal framework -- the cross-tuning set-up is a creative and clean way to isolate bias origins (BmJn, VEAg, mav9)
2. Sound experimental design and statistical analysis (BmJn, VEAg, Eywr, mav9)
3. Accessible writing and effective visuals (BmJn, Eywr)

Weaknesses:
1. All reviewers originally worried that only two base models and two instruction sets might limit generality. The rebuttal addressed this by Mistral-7B-Instruct and Llama-2-13B-Chat.
2. The reviewer found the theory around “latent vs observed” bias a bit fuzzy (mav9, still have some questions after rebuttal)
3. Reviewers found the guideline on mitigation limited and asked for more concrete next steps (e.g., pre-training data filtering, BmJn, mav9). Rebuttal offered some ideas that partially addressed this.
4. Potential dataset-similarity confound, i.e. whether instruction data overlaps with evaluation prompts (VEAg, authors clarified)

Overall, the paper makes good contribution to the field, and I encourage the authors to incorporate their rebuttal responses to the paper revision.